# New Non-Fullerene Acceptor with Extended Conjugation of Cyclopenta [2,1-b:3,4-b’] Dithiophene for Organic Solar Cells

**DOI:** 10.3390/molecules27217615

**Published:** 2022-11-06

**Authors:** Cheng Sun, Sanseong Lee, Changeun Choi, Soyeong Jeong, Juhui Oh, Ju-Hyeon Kim, Jaeyoung Kim, Ho Eon Baek, Hongkyu Kang, Soo-Young Jang, Hyun Ho Choi, Kwanghee Lee, Yun-Hi Kim

**Affiliations:** 1Department of Chemistry and RIGET, Gyeongsang National University, Jinju 52828, Korea; sunchenggnu@gmail.com (C.S.); my_kitten@naver.com (H.E.B.); 2School of Materials Science and Engineering, Gwangju Institute of Science and Technology, Gwangju 61005, Korea; sanseong@gm.gist.ac.kr (S.L.); jumi0504@gm.gist.ac.kr (J.O.); juhyeon-kim@gm.gist.ac.kr (J.-H.K.); jyk941231@gm.gist.ac.kr (J.K.); 3Heeger Center for Advanced Materials, Research Institute for Solar and Sustainable Energies, Gwangju Institute of Science and Technology, Gwangju 61005, Korea; jsy2148@gm.gist.ac.kr; 4Department of Materials Engineering and Convergence Technology and RIGET, Gyeongsang National University, Jinju 52828, Korea; dusrnr3579@naver.com; 5Center for Research Innovation, Gwangju Institute of Science and Technology, Gwangju 61005, Korea; gemk@gist.ac.kr; 6Research Institute for Solar and Sustainable Energies, Gwangju Institute of Science and Technology, Gwangju 61005, Korea

**Keywords:** non-fullerene acceptor, cyclopenta [2,1b:3,4b’] dithiophene, organic solar cell, surface energy, film morphology, ITIC

## Abstract

Herein, we design and characterize 9-heterocyclic ring non-fullerene acceptors (NFAs) with the extended backbone of indacenodithiophene by cyclopenta [2,1-b:3,4-b’] dithiophene (CPDT). The planar conjugated CPDT donor enhances absorption by reducing vibronic transition and charge transport. Developed NFAs with different end groups shows maximum absorption at approximately 790–850 nm in film. Because of the electronegative nature of the end-group, the corresponding acceptors showed deeper LUMO energy levels and red-shifted ultraviolet absorption. We investigate the crystallinity, film morphology, surface energy, and electronic as well as photovoltaic performance. The organic photovoltaic cells using novel NFAs with the halogen end groups fluorine or chlorine demonstrate better charge collection and faster exciton dissociation than photovoltaic cells using NFAs with methyl or lacking a substituent. Photovoltaic devices constructed from m-Me-ITIC with various end groups deliver power conversion efficiencies of 3.6–11.8%.

## 1. Introduction

Organic solar cells (OSCs) are regarded as next-generation photovoltaics because their advantages of lightweight, semi-transparent, tunable energy levels and absorption range, and potential application for flexible devices [1,2,3,4,5,6,7,8]. A typical bulk heterojunction (BHJ) organic solar cell is a combination of acceptor and donor materials. In the early stage of organic solar cells, fullerene derivatives were used as acceptors, owing to their excellent electron and affinity mobility and appropriate nanoscale morphology. [9,10,11,12,13,14] Benefits from the development of device optimization and donor materials [15,16,17,18,19,20], fullerene-based OSCs have exceeded 10% power conversion efficiencies (PCEs) [21,22]. However, limitations such as inferior near-infrared and narrow visible region absorption as well as low tunability of energy level have limited the further development of fullerene derivatives [23,24,25].

To overcome the limitations of fullerene derivatives, ITIC non-fullerene acceptors (NFAs) and their derivatives with acceptor-donor-acceptor (A-D-A) architecture have drawn particular interest owing to broad light absorption, regulable energy levels, stable morphological, and low-cost synthesis potential [26,27,28,29,30,31,32,33]. This A-D-A backbone includes a fused and planar electron-donating unit capped with electron-withdraw end groups. Through manipulation of the backbone conjugated length [34,35,36,37,38], sidechain type and length [39,40,41,42,43], and end groups [44,45], the electronic and morphological properties can easily modify. With the boost of ITIC non-fullerene acceptor, the PCEs of organic solar cells have reached 13–15% [46,47,48,49,50,51,52,53].

In this paper, we utilize extended conjugation: the conjugation of indacenodithiophene is extended by substituting the outward thienothiophene moieties with cyclopenta [2,1-b:3,4-b’] dithiophene (CPDT), which enables construction 9-heterocyclic ring acceptor, meta-methyl-ITIC (m-Me-ITIC). The symmetric and planar conjugated CPDT electron rich moiety enhances absorption because of reduced vibronic transition and enhances the charge transport. Additionally, extended conjugate bridging units improved the absorption in the long-wavelength region. Meta-alkoxy phenyl side chains provide good solubility in common solvents, and different end groups lead to distinct LUMO energy levels with good morphology in the film state [28]. Blend with PBDB-T polymer donor, the m-Me-ITIC device delivers an open-circuit voltage (Voc) of 0.78 V, and short-circuit current density (Jsc) of 23.7 mA·cm^–2^, fill factor (FF) of 64%, and PCE of 11.8%.

## 2. Experimental

### 2.1. Synthesis

#### 2.1.1. Synthesis of Compound **2**

*n*-BuLi (5.4 mL, 13.5 mmol) was dropped into THF solution of 1-bromo-3-((2-ethylhexyl)oxy)benzene (6.0 g, 13.1 mmol) at −78 °C and string for 1.5 h. Compound **1** (2 g, 2.2 mmol) was added to the mixture, and after stirred at 25 °C overnight, the solvent was removed. The crude product was added to 100 mL THF with 10 g amberlyst^®^ 15(H) and stirred for 1 h at 70 °C after cooling down to temperature and removing the solvent, crude purified with silica gel chromatography (hexane/dichloromethane 9:1) to give compound **2** (1.9 g, 65%).

^1^H NMR (300 MHz, CDCl_3_) δ 7.26 (s, 4H), 7.13 (t, *J* = 8.0 Hz, 4H), 7.00 (t, *J* = 2.0 Hz, 4H), 6.88 (d, *J* = 7.9 Hz, 4H), 6.83 (d, *J* = 4.6 Hz, 2H), 6.77 (m, 4H), 3.76 (m, 8H), 1.66 (s, 4H), 1.52–1.18 (m, 8H), 0.91 (s, 12H), 0.86 (t, *J* = 7.3 Hz, 24H).

^13^C NMR (125 MHz, CD_2_Cl_2_) δ 159.54, 157.95, 156.31, 152.45, 151.39, 144.12, 142.83, 140.24, 136.85, 135.18, 129.52, 124.79, 122.21, 121.02, 116.68, 116.60, 115.09, 112.97, 112.88, 70.86, 64.16, 39.83, 37.79, 31.96, 30.89, 29.75, 29.49, 24.58, 24.22, 23.50, 23.08, 14.31, 14.22, 11.30.

#### 2.1.2. Synthesis of Compound **3**

POCl_3_ (1.66 g, 10.8 mmol) was slowly dropped into the mixture of compound **2** (1.8 g, 1.36 mmol) and DMF (0.88 g, 12 mmol) in toluene at −10° C and heated to 75 °C for eight hours. After removing the solvent, the mixture was purified through silica gel column chromatography (hexane/dichloromethane1:3). Compound **3** was obtained as an orange solid (1.5 g, 95%).

^1^H NMR (300 MHz, CD_2_Cl_2_) δ 9.76 (s, 2H), 7.48 (s, 2H), 7.41 (s, 2H), 7.18 (t, *J* = 7.8 Hz, 4H), 6.96 (m, 4H), 6.90 (m, 4H), 6.81 (m, 4H), 3.78 (d, *J* = 5.7 Hz, 8H), 1.81–1.54 (m, 4H), 1.51–1.20 (m, 32H), 0.96 (s, 12H), 0.92–0.78 (m, 24H).

^13^C NMR (125 MHz, CD_2_Cl_2_) δ 182.52, 161.53, 159.75, 158.49, 157.02, 151.54, 147.42, 145.99, 143.76, 143.51, 138.43, 135.78, 129.66, 121.44, 116.58, 115.98, 113.20, 70.92, 63.99, 46.73, 39.71, 30.86, 30.82, 29.44, 24.20, 24.17, 23.92, 23.45, 14.27, 11.25.

#### 2.1.3. Synthesis of M-Me-ITIC Acceptors

INCN, INCN-Me, INCN-F, or INCN-Cl (0.82 mmol) and compound **3** (250 mg, 0.181 mmol) were dissolved in chloroform (30 mL), and pyridine (2 mL) was added. After string overnight at 65 °C, the mixture was poured into CH_3_OH; the solid part was purified through silica-gel column chromatography (hexane/dichloromethane 1:5). m-Me-ITIC acceptors were obtained as blue solid.

m-Me-ITIC, (220 mg, 68%). ^1^H NMR (300 MHz, CD_2_Cl_2_) δ 8.83 (s, 2H), 8.66–8.56 (m, 2H), 7.89–7.80 (m, 2H), 7.70 (dd, *J* = 5.3, 2.4 Hz, 4H), 7.52 (d, *J* = 16.1 Hz, 4H), 7.21 (t, *J* = 8.0 Hz, 4H), 7.02–6.88 (m, 8H), 6.83 (dd, *J* = 8.2, 2.4 Hz, 4H), 3.79 (d, *J* = 5.7 Hz, 8H), 1.66 (m, 4H), 1.49–1.20 (m, 32H), 1.00 (s, 12H), 0.87 (m, 24H). ^13^C NMR (126 MHz, CDCl_3_) δ 188.67, 161.03, 160.73, 159.41, 157.49, 151.68, 142.74, 140.09, 136.89, 136.21, 134.73, 134.02, 129.49, 125.11, 123.38, 121.22, 116.55, 116.35, 115.45, 115.32, 112.95, 70.72, 66.92, 63.67, 46.35, 39.36, 30.58, 30.55, 29.82, 29.17, 29.14, 24.07, 23.91, 23.88, 23.14, 14.22, 11.19, 11.17. MS (MALDI-TOF) calcd. For C_112_H_110_N_4_O_6_S_4_, 1734.7308; found, 1734.7863.

m-Me-ITIC-Me, (210 mg, 66%). ^1^H NMR (300 MHz, CD_2_Cl_2_) δ 8.80 (s, 2H), 8.50 (d, *J* = 8.1 Hz, 0.6H), 8.43 (s, 1.4H), 7.74 (d, *J* = 7.6 Hz, 1.4H), 7.66 (s, 0.6H), 7.59–7.44 (m, 6H), 7.20 (t, *J* = 8.0 Hz, 4H), 6.99–6.87 (m, 8H), 6.82 (dd, *J* = 8.3, 2.3 Hz, 4H), 3.79 (d, *J* = 5.7 Hz, 8H), 2.52 (d, *J* = 3.6 Hz, 6H), 1.66 (p, *J* = 6.1 Hz, 4H), 1.47–1.18 (m, 32H), 0.99 (s, 12H), 0.92–0.80 (m, 24H). ^13^C NMR (125 MHz, CDCl_3_) δ 188.79, 188.47, 160.83, 159.40, 157.41, 151.61, 151.34, 146.23, 145.70, 142.80, 140.49, 137.70, 137.27, 136.53, 136.17, 135.59, 134.99, 134.75, 129.47, 125.55, 125.07, 123.78, 123.31, 121.23, 116.55, 116.30, 115.50, 115.38, 112.95, 70.71, 66.72, 66.27, 63.67, 46.34, 39.36, 30.59, 30.56, 29.17, 29.14, 24.07, 23.91, 23.89, 23.15, 22.63, 22.15, 14.22, 11.20, 11.17. MS (MALDI-TOF) calcd. For C_114_H_114_N_4_O_6_S_4_, 1762.7621; found, 1762.7922.

m-Me-ITIC-F, (250 mg, 78%). ^1^H NMR (300 MHz, CD_2_Cl_2_) δ 8.81 (s, 2H), 8.46 (dd, *J* = 10.3, 6.5 Hz, 2H), 7.66–7.47 (m, 6H), 7.20 (t, *J* = 8.0 Hz, 4H), 6.98–6.87 (m, 8H), 6.82 (dd, *J* = 8.2, 2.4 Hz, 4H), 3.78 (d, *J* = 5.7 Hz, 8H), 1.64 (m, 4H), 1.52–1.20 (m, 32H), 0.99 (s, 12H), 0.95–0.80 (m, 24H).^13^C NMR (125 MHz, CDCl_3_) δ 186.16, 162.93, 161.70, 159.43, 158.49, 157.94, 157.69, 155.23, 155.12, 153.16, 153.06, 153.03, 152.61, 151.76, 142.59, 139.26, 137.63, 137.21, 136.55, 136.33, 134.41, 134.36, 129.50, 121.17, 118.64, 116.58, 116.53, 115.02, 114.94, 114.87, 114.69, 112.96, 112.34, 112.20, 70.72, 67.16, 63.70, 46.37, 39.35, 30.56, 30.53, 29.16, 29.13, 24.01, 23.89, 23.87, 23.13, 14.21, 11.18, 11.15.MS (MALDI-TOF) calcd. For C_112_H_106_F_4_N_4_O_6_S_4_, 1806.6931; found, 1806.7775.

m-Me-ITIC-Cl, (250 mg, 74%). ^1^H NMR (300 MHz, CD_2_Cl_2_) δ 8.81 (s, 2H), 8.66 (s, 2H), 7.75 (d, *J* = 2.0 Hz, 2H), 7.54 (d, *J* = 3.1 Hz, 4H), 7.19 (t, *J* = 8.0 Hz, 4H), 6.96–6.88 (m, 8H),6.84–6.76 (m, 4H), δ 3.77 (d, *J* = 5.7 Hz, 8H),1.63 (m, 4H), 1.48–1.19 (m, 32H), 0.99 (s, 12H), 0.86 (m, 24H).^13^C NMR (125 MHz, CDCl_3_) δ 186.11, 163.07, 161.94, 159.41, 158.56, 158.29, 157.78, 153.05, 151.80, 142.55, 139.66, 139.42, 138.91, 138.70, 137.95, 137.33, 136.40, 135.96, 129.49, 126.58, 124.67, 121.18, 118.54, 116.72, 116.53, 115.12, 114.94, 112.92, 70.75, 67.02, 63.71, 46.35, 39.34, 30.55, 30.52, 29.16, 29.12, 24.00, 23.87, 23.14, 14.22, 11.19, 11.16. MS (MALDI-TOF) calcd. For C_112_H_106_Cl_4_N_4_O_6_S_4_, 1870.5749; found, 1870.6914.

## 3. Result and Discussion

The synthetic scheme of m-Me-ITIC series was presented in Figure 1. We treated compound **1** with freshly prepared (3-((2-ethylhexyl)oxy)phenyl)lithium-yielded diol, then intramolecular Friedel–Crafts cyclization to was carried out to furnish compound **2** (IT). This IT was reacted with POCl_3_ and dimethylformamide to yield carbaldehyde, which was then capped with various IC groups by the Knoevenagel condensation reaction to generate the desired compound. Each of the new acceptors was characterized by NMR, and mass spectrometry (Appendix A). The resulting acceptors showed good solubility at room temperature in common solvent.

Thermogravimetric analysis (TGA) and differential scanning calorimetry (DSC) was carried out to investigate the thermal stability of the m-Me-ITIC acceptors (Appendix A). All four acceptors have decomposition temperatures (Td) > 330 °C (5% weight loss), indicating m-Me-ITIC series have good thermal stability. Differential scanning calorimetry revealed that m-Me-ITIC-Cl undergoes an endothermic melting transition at 144 °C. This thermal transition implies that m-Me-ITIC-Cl has a greater tendency to crystallize than the other derivatives because of end group differences.

The ultraviolet–visible absorption spectra and corresponding data of the m-Me-ITIC series in chloroform and thin film are shown in Figure 1b,c and Table 1. The entire series exhibited large red-shifts > 100 nm compared with ITIC-ethylhexyl oxy (OEH), which was composed of thienothiophene instead of CPDT due to the extended conjugation. Furthermore, measurement of ultraviolet–visible spectra in solution revealed that λ_max_ exhibited greater red-shifts with increasing electron acceptor ability: m-Me-ITIC-Me (λ_max_: 756 nm) < m-ITIC (760 nm) < m-ITIC-F (775 nm) < m-ITIC-Cl (790 nm). In thin film, a broader spectrum and redshifted maximum absorption were observed for m-Me-ITIC acceptors; the absorption maximum of m-Me-ITIC-Cl was shifted by nearly 16 nm compared with m-Me-ITIC-F, suggesting that stronger π–π interactions occur in thin film. We conducted our electrochemical study of the m-Me-ITIC series using cyclic voltammetry for energy level measurement (Appendix A). Owing to the electronegative nature of INIC group, the LUMO energy levels of the corresponding NFAs decreased from −4.19 and −4.20 eV for m-Me-ITIC-M and m-Me-ITIC, respectively, to −4.29 eV for m-Me-ITIC-F and −4.35 eV for m-Me-ITIC-Cl.

To demonstrate the photovoltaic performances of m-Me-ITIC acceptors, we fabricated organic photovoltaic (OPV) devices by blending m-Me-ITIC acceptors with PBDB-T polymer donor using an inverted structure with glass/tin-doped indium oxide/ZnO/BHJ/MoO_x_/Ag. Current density–voltage (*J–V*) characteristics of the optimized devices with the m-Me-ITIC acceptors was measured under simulated AM1.5 G with an intensity of 100 mW/cm^2^, the results are shown in Figure 2a and the device data are summarized in Table 2. To verify the measurement accuracy, we crosschecked the external quantum efficiency spectra with the *J–V* characteristics (Figure 2b and Table 2). We summarize the device optimization data (e.g., thermal annealing, donor/acceptor blending ratio, and processing additive of each OPV device) (Appendix A). Thus, the efficiency-optimized devices made from m-Me-ITIC, m-Me-ITIC-Me, m-Me-ITIC-F, and m-Me-ITIC-Cl demonstrated PCE values of 5.9%, 3.6%, 11.8%, and 10.8%, respectively.

To further understand the collection behaviors and charge dissociation of the four fabricated OPV cells, the photocurrent density (*J_ph_*) on the effective voltage (*V_eff_*) was investigated, the results are presented in Figure 2c. *V_eff_* was determined using the following equation: 
$V_{eff}=V_{0}-V_{a}$
, where *V*_0_ indicates the voltage when illuminated with a current density (*J_L_*) identical to the dark current density (*J_D_*), and *V_a_* indicates the applied voltage. *J_ph_* was determined using the following equation: 
$J_{ph}=J_{L}-J_{D}$
 [52,53]. Each of the OPV cells was biased from −2.0 to 1.0 V to demonstrate the exciton dissociation process; the saturation current density (*J_sat_*) was verified for a *V_eff_* of 2.5 V. The *J_sat_* values measured for m-Me-ITIC, m-Me-ITIC-Me, m-Me-ITIC-F, and m-Me-ITIC-Cl were 14.12, 9.20, 24.00, and 22.40 mA·cm^−2^. We divided the normalized *J_ph_* by *J_sat_* to calculate P(E,T) to demonstrate the exciton dissociation as well as charge collection properties. The P(E,T) values calculated were 79.00%, 71.24%, 96.84%, and 94.08%, respectively, indicating that OPV cells using novel NFAs with the halogen end groups fluorine or chlorine demonstrate better charge collection and faster exciton dissociation than OPV cells using the other two NFAs [54].

To investigate the charge transport behaviors, we analyzed the *J_sc_* value of each cell by varying the irradiated light intensity (*P_light_*) from 0.93 to 100 mW·cm^−2^, as shown in Figure 2d. The relationship between *J_sc_* and *P_light_* can be described as 
$J_{sc}\propto P_{light}^{\alpha }$
, where *α* is an exponential factor. A value for *α* close to unity indicates weak bimolecular recombination. As the value of *α* decreases, severe bimolecular recombination occurs [55]. The *α*-values of m-Me-ITIC-F blend and m-Me-ITIC-Cl blend was calculated as 0.9659 and 0.9673, which is higher than the *α*-values of m-Me-ITIC blend and m-Me-ITIC-Me blend (0.9519 and 0.9526, respectively). Because bimolecular recombination induces partial charge carrier losses [56], we presume that minimizing bimolecular recombination will help to increase the *J_sc_* value and fill factor of the OPV devices based on NFAs with halogen end groups (m-Me-ITIC-F and m-Me-ITIC-Cl).

To study the charge transport properties of the OSCs further, the devices’ electron and hole mobilities were evaluated by the space charge limited current (SCLC) method (Appendix A). The electron mobility and hole mobility were measured with the device structure of ITO/ZnO/Active layer/BCP/Al and ITO/PEDOT:PSS/Active layer/MoOx/Ag, respectively. The calculated hole/electron (
$\mu_{h}/\mu_{e}$
) of the PBDB-T:m-Me-ITIC, PBDB-T:m-Me-ITIC-Me, PBDB-T:m-Me-ITIC-F, and PBDB-T:m-Me-ITIC-Cl devices were 
$4.38\times 10^{-4}/5.62\times 10^{-4}$
, 
$4.46\times 10^{-4}/5.53\times 10^{-4}$
, 
$6.55\times 10^{-4}/6.47\times 10^{-4}$
, and 
$5.17\times 10^{-4}/5.61\times 10^{-4}\;{\mathrm{cm}}^{2}\;V^{-1}\;s^{-1}$
, which is equivalent to 
$\mu_{h}/\mu_{e}$
 ratios of 0.78, 0.81, 1.01, and 0.92, respectively. The PBDB-T:m-Me-ITIC-F and PBDB-T:m-Me-ITIC-Cl devices showed higher and more balanced charge transport than PBDB-T:m-Me-ITIC, PBDB-T:m-Me-ITIC-Me, which could reduce charge recombination, thus contributing to the enhanced *J_sc_* and fill factor.

Atomic force microscopy (AFM) was employed to investigate the morphologies of the m-Me-ITIC-, m-Me-ITIC-Me-, m-Me-ITIC-F-, and m-Me-ITIC-Cl-based BHJ films, the detail was shown in Figure 3. The root-mean-square surface roughness (RMS) values were higher for all BHJ blend films than for the novel NFA films. The RMS values of m-Me-ITIC and m-Me-ITIC-Me blend were <1 nm (0.97 and 0.83 nm, respectively), while the values of m-Me-ITIC-F and m-Me-ITIC-Cl blend were close to 1.5 nm. The exciton dissociation, charge transport, and charge collection ability of OPVs can be determined by BHJ film morphology. Favorable phase separation between donor and acceptor in m-Me-ITIC-F blend and m-Me-ITIC-Cl blend may have produced a slightly rougher surface, increasing the efficiency of these OPV cells [57,58]. These results consistently show the effects of the halogen end group on the novel NFAs.

To confirm the crystallinity of the fabricated BHJ films based on novel NFAs, X-ray diffraction was performed. Figure 4a shows the X-ray diffraction patterns of neat acceptor films. 2θ diffraction peaks are located at 5.22°, 4.20°, 3.54°, and 3.39°, respectively; these values correspond to d-spacings of 16.91, 21.02, 24.94, and 26.02 Å, respectively. Notably, the diffraction peaks of the m-Me-ITIC-F and m-Me-ITIC-Cl films are clear and distinct. As shown in Figure 4b, the 2θ diffraction peaks of m-Me-ITIC, m-Me-ITIC-F, and m-Me-ITIC-Cl BHJ blend films are located at 3.20°, 3.61°, and 3.66°, respectively; these values correspond to d-spacings of 27.61, 24.46, and 24.09 Å, respectively. When the novel NFAs were mixed with the donor to produce blended BHJ films, the crystallinity of the PBDB-T:m-Me-ITIC-Me film almost disappeared compared to pristine m-Me-ITIC-Me. Although the peaks of the other BHJ films remained visible, lamellar distances were shorter for m-Me-ITIC-F blend and m-Me-ITIC-Cl blend than m-Me-ITIC blend. This might be caused by strong intermolecular interactions through the highly electronegative halogen atoms used as end groups, which could induce greater molecular polarization [59]. This might be consistent evidence for highly efficient OPV cells based on m-Me-ITIC-F and m-Me-ITIC-Cl.

## 4. Conclusions

We designed and synthesized a 9-heterocyclic ring acceptor, m-Me-ITIC, with different end groups by substituting the outward thienothiophene moieties with CPDT at indacenodithiophene. The developed m-Me-ITIC, m-Me-ITIC-Me, m-Me-ITIC-F, and m-Me-ITIC-Cl showed long-wavelength absorption at approximately 756–790 nm in solution because of the symmetric, planar, and extended conjugated CPDT donor. The m-Me-ITIC-based acceptors were developed into polymer solar cells that contained benzodithiophene-based donor polymers. The devices optimized for efficiency demonstrated PCEs of 5.9%, 3.6%, 11.8%, and 10.8% using m-Me-ITIC, m-Me-ITIC-Me, m-Me-ITIC-F, and m-Me-ITIC-Cl, respectively. Analyses of the charge dissociation and collection behaviors of the four fabricated OPV cells found that the new NFAs with halogen end groups showed salient bimolecular recombination dynamics. The BHJ film morphology for exciton dissociation, charge transport, and charge collection ability of the OPVs showed favorable phase separation in m-Me-ITIC-F blend and m-Me-ITIC-Cl blend. The shorter lamellar distances of these two materials, caused by strong intermolecular interactions through the highly electronegative halogen atoms, could induce greater molecular polarization. This may also be consistent evidence for highly efficient OPV cells based on m-Me-ITIC-F and m-Me-ITIC-Cl.

## Data Availability

Not applicable.

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
