# Peer review of "New Non-Fullerene Acceptor with Extended Conjugation of Cyclopenta [2,1-b:3,4-b’] Dithiophene for Organic Solar Cells"

_molecules, 2022, doi:10.3390/molecules27217615_

Round 1

Reviewer 1 Report

The manuscript by Sun et al. presented new non-fullerene acceptor of m-Me-ITIC, m-Me-ITIC-Me, m-Me-ITIC-F, and m-Me-ITIC-Cl for organic solar cells. It is well organized and can be accepted after addressing following points.

1.    The description of Y-series acceptors in “Introduction” section may be not suitable for your theme.

2.    “Scheme 1. Synthetic scheme for m-Me-ITIC acceptors” is not correct. Revised as “Scheme 1. Synthetic scheme for m-Me-ITIC-series acceptors.”

3.    Please present the absorption coefficient of m-Me-ITIC-series acceptors.

4.    Please present the electron/hole mobility of blend films.

5.    N-BuLi should revised as n-BuLi; pyridine (2 ml) should revised as pyridine (2 mL); current density (JL) identical to the dark current density (JD) should revised as JL and JD.

Author Response

Response to the Reviewer’s COMMENTS:

Reviewer 1

The manuscript by Sun et al. presented new non-fullerene acceptor of m-Me-ITIC, m-Me-ITIC-Me, m-Me-ITIC-F, and m-Me-ITIC-Cl for organic solar cells. It is well organized and can be accepted after addressing following points.

  1. The description of Y-series acceptors in “Introduction” section may be not suitable for your theme.

Respond) Thank you for valuable comments. We revised the introduction in the manuscript according referee’s comments.

In manuscript, the paragraph was removed “Another breaking ~ high efficiency”

  1. “Scheme 1. Synthetic scheme for m-Me-ITIC acceptors” is not correct. Revised as “Scheme 1. Synthetic scheme for m-Me-ITIC-series acceptors.”

Respond) Thank you for your kind comments. We revised the cation of synthetic schem1.

Scheme 1. Synthetic scheme for m-Me-ITIC series acceptors.

  1. Please present the absorption coefficient of m-Me-ITIC-series acceptors.

Respond) Thank you for valuable comments. We added the absorption coefficient of m-ITIC series acceptors in Table 1.

  1. Please present the electron/hole mobility of blend films.

Respond) Thank you for your constructive comments. We conducted SCLC analysis to gain the electron/hole mobility of blend films. SCLC method is a widely used analysis method for obtaining the charge mobility in the field of organic solar cells.
 Our results showed that there is a meaningful trend in the hole/electron mobility between the four blend films. We have inserted this result in text, and supporting information (Figure S10 and Table S7). The contents are as follows.

To study the charge transport properties of the OSCs further, the devices' electron and hole mobilities were evaluated by the space charge limited current (SCLC) method (Figs. S10 and Table S7 in Supporting Information). The electron mobility and hole mobility were measured with the device structure of ITO/ZnO/Active layer/BCP/Al and ITO/PEDOT:PSS/Active layer/MoOx/Ag, respectively. The calculated hole/electron () of the PBDB-T:m-Me-ITIC, PBDB-T:m-Me-ITIC-Me, PBDB-T:m-Me-ITIC-F, and PBDB-T:m-Me-ITIC-Cl devices were , , , and , which is equivalent to  ratios of 0.78, 0.81, 1.01, and 0.92, respectively. The PBDB-T:m-Me-ITIC-F and PBDB-T:m-Me-ITIC-Cl devices showed higher and more balanced charge transport than PBDB-T:m-Me-ITIC, PBDB-T:m-Me-ITIC-Me, which could reduce charge recombination, thus contributing to the enhanced Jsc and fill factor.

  1. N-BuLi should revised as n-BuLi; pyridine (2 ml) should revised as pyridine (2 mL); current density (JL) identical to the dark current density (JD) should revised as JLand JD.

   Respond) Thank you for kind comments. We revised the comments.

In the manuscript, “n-BuLi (5.4 mL, 13.5 mmol) ~was removed”

“INCN, INCN-Me, INCN-F, or INCN-Cl (0.82 mmol) and compound 3 (250 mg, 0.181 mmol) were dissolved in chloroform (30 mL), and pyridine (2 mL) was added.”

“Veff was determined using the following equation: , where V0 indicates the voltage when illuminated with a current density (JL) identical to the dark current density (JD), and Va indicates the applied voltage.”

Reviewer 2 Report

The authors synthesized four small molecules based on cyclopenta[2,1-b:3,4-b’]dithiophene. The result of power conversion efficiencies are fair but are among the top combination of polymers and ADA-type small molecules. The development and optimization of new small molecules might benefit the OPV community. Nonetheless, several concerns must be addressed and altered to support the conclusion of high PCEs. We feel that the suggested measurements will aid in providing a more comprehensive conclusion to support the result of presenting excellent cyclopenta[2,1-b:3,4-b’]dithiophene derivatives. I would recommend it be published in Molecules after the following issues are addressed:

1. Some Subscripts and superscripts are not consistent in the whole manuscript and Supplementary Information; please check those typos before submission. (For example, ln 208, p. 6; SI ln 21, p. 1)

2. Please add the Fc/Fc+ reference of cyclic voltammetry measurements to the Supporting Information.

3. Please measure the absorption coefficient of binary blend and pure synthesized material films for UV−vis absorption spectra to precisely discuss the differences between curves since the Jsc value is the main factor in increasing the PCE in this work. In addition, please correct the label of Figure 1c.

4. In BHJ OPVs, the miscibility between materials is of importance. Please measure the contact angle of each pure material and verify the origin of high PCEs other than just the PCE itself. (check J. Mater. Chem. A, 2021, 9, 20510-20517 for the measurement)  

5. For the introduction, several works of origin of ITIC, meta-positioned alkoxy chain, and chlorinated end group should be cited. (Adv.Mater.2015, 27, 1170–1174; J. Mater. Chem. A, 2019, 7, 3072-3082; Sol. RRL2020,4, 2000253; Adv.Mater.2021, 33, 2006120)

6. In ln 57, p.2, it should be thienothiophene instead of thiophene.

7. In ln 59, p.2, please change the word from donor to electron-rich moiety so the reader will understand.

8. Since this work is mainly about incorporating cyclopenta[2,1-b:3,4-b’]dithiophene into NFA, can the authors add more background on why they use this moiety and the property of this moiety?

9. The description of Y-series molecules should be addressed since the Y1 molecule (using benzotriazole) is the origin of Y-series high PCE small molecule acceptors. The authors could check ACS Nano 2021, 15, 12, 18679–18682 for more information.

10. How many devices are tested for each kind of condition? The error bar with the correct effective number should be included in the manuscript and supplementary information.

Author Response

Reviewer 2

The authors synthesized four small molecules based on cyclopenta[2,1-b:3,4-b’]dithiophene. The result of power conversion efficiencies are fair but are among the top combination of polymers and ADA-type small molecules. The development and optimization of new small molecules might benefit the OPV community. Nonetheless, several concerns must be addressed and altered to support the conclusion of high PCEs. We feel that the suggested measurements will aid in providing a more comprehensive conclusion to support the result of presenting excellent cyclopenta[2,1-b:3,4-b’]dithiophene derivatives. I would recommend it be published in Molecules after the following issues are addressed:

1. Some Subscripts and superscripts are not consistent in the whole manuscript and Supplementary Information; please check those typos before submission. (For example, ln 208, p. 6; SI ln 21, p. 1)

Respond) Thank you for kind comments. We revised the typos

  1. Please add the Fc/Fc+ reference of cyclic voltammetry measurements to the Supporting Information.

Respond) Thank you for kind comments. We added the Fc/Fc+ reference of cyclic voltammetry measurements.

Figure S9. CV curves measured from Chloroform solution of m-Me-ITIC-Xs.

  1. Please measure the absorption coefficient of binary blend and pure synthesized material films for UV−vis absorption spectra to precisely discuss the differences between curves since the Jsc value is the main factor in increasing the PCE in this work. In addition, please correct the label of Figure 1c.

Respond) Thank you for the valuable comments. We measured the absorption coefficient of synthesized materials in the Table 1. The absorption coefficient of synthesized m-ITIC derivatives are similar. Therefore, we think minimized bimolecular recombination caused by suitable nanoscale morphology (Figure 3) and increased and well-balanced hole/electron mobility (Table S7 added in supporting information) are main factors of enhancing Jsc on the results of the experiment.

  1. In BHJ OPVs, the miscibility between materials is of importance. Please measure the contact angle of each pure material and verify the origin of high PCEs other than just the PCE itself. (check J. Mater. Chem. A, 2021, 9, 20510-20517 for the measurement)  

Respond) Thank you for your constructive comments. We measured the contact angles of PBDB-T and the four acceptors to gain the difference in surface energies. Then, we wanted to figure out the miscibility through the Flory-Huggins equation. However, the contact angles of the four acceptor films were almost the same within the 1% range (m-Me-ITIC : 91.13°, m-Me-ITIC-Me : 89.92°, m-Me-ITIC-F : 90.91°, m-Me-ITIC-Cl : 92.01°). We thought it is not suitable for explaining the origin of the difference in PCE.

  1. For the introduction, several works of origin of ITIC, meta-positioned alkoxy chain, and chlorinated end group should be cited. (Adv.Mater.2015, 27, 1170–1174; J. Mater. Chem. A, 2019, 7, 3072-3082; Sol. RRL2020,4, 2000253; Adv.Mater.2021, 33, 2006120)

Respond ) Thank you for kind comments. We added the references.

  1. In ln 57, p.2, it should be thienothiophene instead of thiophene.

Respond ) Thank you for kind comments. We revised the thiophene to thienothiophene.

In the manuscript, “we utilize extended conjugation: the conjugation of indacenodithiophene is extended by substituting the outward thienothiophene moieties with cyclopenta [2,1‑b:3,4‑b’]dithiophene (CPDT)”

  1. In ln 59, p.2, please change the word from donor to electron-rich moiety so the reader will understand.

Respond ) Thank you for kind comments. We revised donor to electron rich moiety

In the manuscript, The symmetric and planar conjugated CPDT electron- rich moiety enhances absorption because of reduced vibronic transition and charge transport

  1. Since this work is mainly about incorporating cyclopenta[2,1-b:3,4-b’]dithiophene into NFA, can the authors add more background on why they use this moiety and the property of this moiety?

Respond ) Thank you for kind comments. As we described in the introduction part, the symmetric and planar conjugated CPDT electron rich moiety enhances absorption because of reduced vibronic transition and enhanced charge transport. Additionally, extended conjugate bridging units improved the absorption in the long-wavelength region.

  1. The description of Y-series molecules should be addressed since the Y1 molecule (using benzotriazole) is the origin of Y-series high PCE small molecule acceptors. The authors could check ACS Nano 2021, 15, 12, 18679–18682 for more information.

Respond ) Thank you for kind comments. As suggestion of reviewer 1, we deleted the paragraph of the description of Y-Series in the Introduction

  1. How many devices are tested for each kind of condition? The error bar with the correct effective number should be included in the manuscript and supplementary information.

Respond) Thank you for your comments. In the optimizing process, we tested each condition with two devices (Table S1-S6 in supporting information). And we averaged the each optimized condition with 10 devices. Also, we think that standard deviation can also be a good way of demonstrating the reliability of the device performance. Therefore, we added the standard deviation values to Table 2 in the manuscript.